# Prevalence of Food Allergy in France up to 5.5 Years of Age: Results from the ELFE Cohort

**DOI:** 10.3390/nu14173624

**Published:** 2022-09-02

**Authors:** Sarah Tamazouzt, Karine Adel-Patient, Antoine Deschildre, Caroline Roduit, Marie Aline Charles, Blandine de Lauzon-Guillain, Amandine Divaret-Chauveau

**Affiliations:** 1Pediatric Allergy Department, Children’s Hospital, University Hospital of Nancy, 54511 Vandoeuvre-lès-Nancy, France; 2Paris-Saclay University, CEA, INRAE, DMTS, 91190 Gif-sur-Yvette, France; 3CHU Lille, Pediatric Pulmonology and Allergy Unit, Lille University, Jeanne de Flandre Hospital, 59000 Lille, France; 4Christine Kühne Center for Allergy Research and Education (CK-CARE), 7265 Davos, Switzerland; 5Children’s Hospital, University of Zürich, 8032 Zürich, Switzerland; 6Children’s Hospital of Eastern Switzerland, 9000 St. Gallen, Switzerland; 7Paris Cité University, Inserm, INRAE, CRESS, 75000 Paris, France; 8Joined Unit Inserm-Ined-EFS Elfe, INED, 93300 Aubervilliers, France; 9EA3450 DevAH, University of Lorraine, 54511 Vandoeuvre-lès-Nancy, France

**Keywords:** atopy, children, epidemiology, food allergy, prevalence

## Abstract

Background: In France, updated data on food allergies (FAs) are lacking, despite the need for efficient FA management and prevention. This study aimed to evaluate the prevalence of FAs in children in France, describe the most common allergens and determine the prevalence of atopic diseases in children with FAs. Methods: The ELFE study comprises a French nationwide birth cohort, including 18,329 children born in 2011. FAs were assessed by parental reports of food avoidance based on medical advice related to FAs, provided at 2 months and 2, 3.5 and 5.5 years of age. Data regarding FAs were available for 16,400 children. Data were weighted to account for selection and attrition bias. Results: From birth to 5.5 years of age, FAs were reported for 5.94% (95% CI: 5.54–6.34) children. Milk was the most common allergen, followed by egg, peanut, exotic fruits, tree nuts, gluten and fish. Among children with FAs, 20.5% had an allergy to at least two different groups of allergens; 71% reported eczema at least once before 5.5 years of age; 24.4% reported incidence of asthma; and 42.3% reported incidence of allergic rhinitis or conjunctivitis. Conclusion: In France, the prevalence of FAs in children up to 5.5 years of age is approximately 6%. It was demonstrated that 1 in 5 children with allergies had multiple FAs.

## 1. Introduction

As a part of the “Atopic march”, food allergies (FAs) are considered to comprise the “second wave” of the allergy epidemic after respiratory allergies [1], with several studies evidencing an increase in FA prevalence [2,3] and severity [4].

In 2014, a meta-analysis [2] reported heterogenous FA prevalence in Europe, ranging from 0.1% to 6%. More recently, the European birth cohort EUROPREVALL estimated the prevalence of FAs in school-aged children at 1.4–3.8% [5]. Regarding French data, in 1999, a study of 6672 children aged 9–11 years recruited in six French cities reported an FA prevalence of 2.1% [6]. In 2002, a school survey study including 2716 children aged 2–14 years from eight schools in Toulouse reported a prevalence of 4% [7]. In 2014–2015, the Third National Individual Study of Food Consumption, including 2084 children aged 0–17 years, estimated an FA prevalence of 4.2% [8].

Data on the prevalence and identification of the most common allergens and related allergic diseases are important for clinicians to prevent and adapt the management of FAs. Such data are particularly useful in the development of infant feeding guidelines. Indeed, early introduction of peanut (from 4 to11 months) in atopic children was associated with a reduced prevalence of peanut allergy at 5 years of age [9]. The EAT study showed the same trend for peanut and egg allergies in the general population, but families may have difficulties introducing several food allergens in early life [10]. Awareness of the most frequent food allergens would help in targeting prevention recommendations regarding early introduction. Finally, knowledge of the most common allergens can help to optimize allergen labeling regulations, to limit allergic accidents.

This study aimed to evaluate the prevalence of FAs in French children from birth to 5.5 years of age, describe the most common allergens involved in FAs and determine the prevalence of atopic diseases in children with FAs.

## 2. Materials and Methods

### 2.1. Study Design

The present study was based on data collected within the “Etude Longitudinale Française depuis l’Enfance” (ELFE) study, a French nationwide birth cohort including 18,329 children born in a random sample of maternity units in mainland France in 2011 [11]. Parents reported their child’s health data during phone interviews conducted at 2 months, 1 year, 2 years, 3.5 years and 5.5 years of age. Each interview covered the period since the last follow-up.

Participating mothers had to provide written consent for their own and their child’s participation. Fathers signed the consent form for the child’s participation when present at inclusion or were informed about their rights to oppose it. The ELFE study was approved by the Advisory Committee for Treatment of Health Research Information (10.623), the National Data Protection Authority (910504) and the National Statistics Council (2011X716AU).

### 2.2. Food Allergies

Parents reported any medical diagnosis of cow’s milk allergy at the 2-month follow-up. At the 2-, 3.5- and 5.5-year follow-ups, parents reported any medical advice to avoid certain foods because of an FA. The concerned allergens were reported in accordance with a predefined list (milk, egg, gluten, peanut, fish, exotic fruits (kiwi, pineapple) and soy), as well as using an open-ended question. All allergens reported through this open-ended question were then classified into eight additional groups (Table 1). Among the answers to the open-ended question, answers concerning food intolerance (lactose intolerance), foods with high histamine content (tomato, chocolate, strawberry, raspberry, pork meat products) and food avoidance for a reason other than a child’s allergy (i.e., as a precaution or for other illness) were not considered as an FA. No information regarding symptoms or diagnosis procedure was obtained from the parents.

The allergen groups were established at each follow-up. Children with at least one reported FA between 0 and 5.5 years of age were used to define FA according to the specific allergen (i.e., milk, gluten, egg, etc.) or group of allergens (i.e., tree nuts (TNs), sea food, legumes). If a food avoidance due to a FA to this allergen or group of allergens was reported during at least one follow-up, the child was considered allergic. If a food avoidance due to a FAwas reported for more than two allergens or groups of allergens during the 0–5.5-year period, children were considered as having multiple-group FAs, even if the avoidance was not reported at the same follow-up.

### 2.3. Other Data

Children were considered to have eczema if parents reported at least one incidence of an itchy rash (2 months, 1 year) or a medical diagnosis of eczema (1, 2, 3.5 and 5.5 years). Children were considered to have asthma if a medical diagnosis of asthma was reported at least once [12]. At the 5.5-year follow-up, parents also reported any incidence of allergic rhinitis or conjunctivitis (ARC).

At the 2-month interview, family history of atopy was collected, including parents’ and siblings’ asthma, eczema and allergic rhinitis. Children were considered to have a family history of atopy if at least one parent or sibling had asthma, eczema or allergic rhinitis.

At the 2-year follow-up, a questionnaire was completed by the physician that included items on FAs and respiratory allergies. In cases of allergies, the physician specified the method used for diagnosis, if any (IgE or skin prick tests). Data on the use of the oral food challenge (OFC) procedure was not collected. This questionnaire was completed for 7557 children.

### 2.4. Sample Selection

The study excluded infants whose parents withdrew consent during the first year (*n* = 57). All children with at least one follow-up from 2 months to 5.5 years of age were included (*n* = 16,400).

### 2.5. Statistical Analysis

To provide national statistics on food allergy (FA) prevalence, data were weighted to account for the inclusion procedure and biases related to non-consent [13]. Weighting also included calibration on margins from the state register’s [14] statistical data and the 2010 French National Perinatal study on the following variables: age, region, marital status, migration status, level of education and primiparity. This weighting was calculated for the subsample that completed each follow-up (https://www.elfe-france.fr/fichier/rte/178/Coté%20recherche/Weighting-Elfe-surveys-general-document.pdf, accessed on 1 December 2019) and was used for calculating each prevalence, except those calculated only on the subsample of children with FAs.

The agreement between parent- and physician-reported FAs was evaluated using the Cohen kappa coefficient.

The chi-squared test was used to compare the prevalence of family atopy in children with multiple- or single-group FAs and allergy reporting between physician and parents.

All analyses involved using SAS 9.4 (SAS Institute, Cary, NC, USA). *p* < 0.05 was considered statistically significant.

## 3. Results

The sample characteristics are described in Table 2.

### 3.1. Prevalence of Parent-Reported Food Allergy

Parents reported a doctor-diagnosed FA for 5.94% (95% CI: 5.54–6.34) of children up to 5.5 years of age (Table 3). The most frequently reported allergen was milk, followed by egg, peanut, exotic fruits (such as kiwi and pineapple) and TNs.

From 2 to 5.5 years of age, the overall prevalence of FA tended to decrease: rates of 3.40% (3.01–3.80) at age 2 years, 3.17% (2.79–3.56) at 3.5 years and 2.71% (2.33–3.09) at 5.5 years were reported. Milk was the most common allergen at each follow-up, but its prevalence decreased between 2 months and 5.5 years of age (Figure 1). The prevalence of peanut or TN allergy was higher at 5.5 years than at 2 years of age, whereas an opposite temporal trend was observed for egg and fish.

Multiple-group FAs were found for 1.13% (0.96–1.30) of children (corresponding to 20.5% of children with an FA). The most frequently associated allergens were peanut, milk and egg (Figure 2).

Milk was the most frequently reported allergen for those with only one FA, but 16.8% of children with a milk allergy had at least one other FA (Table 4). In contrast, most children with a soy or mustard allergy had multiple-group FAs, and the associated allergens were milk (64.3 and 40.0%, respectively), peanut (50.0 and 33.3%), egg (35.7 and 73.3%) and TNs (42.9 and 26.7%). For children allergic to peanut, 28.4%, 25.4% and 3.0% had concomitant FAs to egg, TNs or legumes, respectively. For children allergic to TNs, 42.5% were also allergic to peanut and 22.5% to egg. Among children with fish allergy, 10.7% of children were also allergic to seafood/shellfish.

### 3.2. Prevalence of Physician-Reported Food Allergy

At the 2-year follow-up, the prevalence of physician-reported FAs was 3.14% (95% CI: 2.61–3.67) versus 3.40% (3.01–3.80) for parent-reported FAs. Only 43.6% of FAs were confirmed by specific IgE assays or skin prick tests (25.3% specific IgE assay only; 34.5% skin prick tests only; 42.2% both). The agreement between parent- and physician-reported FAs was moderate (kappa = 0.50).

### 3.3. Family History of Atopy and Allergic Diseases

For children with a family history of atopy, the cumulative prevalence of FAs was 7.8%, compared with 4.4% for children without any family history of atopy. Among children with reported FAs, 24.4% also had asthma, 71.0% had eczema and 42.3% had ARC (vs. 12.6%, 44.4% and 30.9%, respectively, for children without FA). The overlap of FAs, eczema and asthma is represented in Figure 3.

## 4. Discussion

In the first French nationwide birth cohort, including children born in 2011, the prevalence of parent-reported FAs in children up to 5.5 years of age was 5.94%. The main food allergens reported were milk, egg, peanut, exotic fruits and TNs. Milk was the most frequently reported single FA. As a whole, 20.5% of children with an FA had an allergy to at least two different groups of allergens. Many children with FA had eczema, and 24% had asthma.

Despite the fact that these are the first data gathered from a representative cohort, the prevalence of FAs has been estimated at 2.1–7.4% in previous publications [6,7,8,9]; therefore, this would suggest that there has been no significant increase within the last 20 years. The estimated prevalence in our study agrees with other studies conducted in Western countries: the prevalence of self-reported FAs in US children was 6.5% in the National Health and Nutrition Examination Survey 2007–2010 [15], 8% in a large US cohort in 2010 [16] and 6.2% in Dutch teenagers [17]. FA prevalence is known to vary by children’s age [7,16]. When comparing the current results on age range, the estimated prevalence of parent-reported FAs at 3.5 years of age in the ELFE study was similar to the prevalence of 3.8% estimated by an oral food challenge (OFC) in the Australian HealthNuts study at 4 years of age [18]. In the Isle of Wight birth cohort, the symptom-based prevalence of FAs was 4.4% at 2 years and 5.0% at 4 years of age, which were slightly higher than this study’s estimates [19]. Different diagnostic criteria of allergy can also induce variations in FA prevalence estimations. In the EuroPrevall cohort, the prevalence of FAs in school-aged children (aged 6–10 years) varied according to the strategy for allergy diagnosis (1.4% with OFC vs. 3.8% with parent-reported allergies) and weighting attrition [5].

Among the allergens involved in FAs in the ELFE study, the most frequently occurring were milk, egg and peanut, as in other studies [7,16,18,19,20]. The prevalence of milk and egg allergies was consistent with that in other studies using reported cases. In a recent review, the prevalence of cow’s milk allergy (CMA) was estimated at 0.5–3% at 1 year of age, depending on the method used to identify allergies (OFC or parental report) [21]. In the EuroPrevall cohort, the prevalence of CMA, as confirmed by an OFC, was 0.54% at 2 years old versus 1.38% at 2 years old in our cohort [22]. In the Isle of Wight birth cohort, the prevalence of CMA, based on reported IgE-mediated symptoms between ages of 1 and 4 years, varied from 1.6% to 3.5%, with the highest prevalence at 1 year of age [19].

Regarding egg allergy, our findings are consistent with those from the Isle of Wight cohort (1.1–1.4% for children aged 1–4 years) [19] and those from the EuroPrevall cohort (1.23% in the first 2 years of life) [23]. In the HealthNuts Study, the prevalence of egg allergy, confirmed by a positive OFC to raw egg, was high (9.5% at 1 year old). This difference is probably due to the use of raw egg, the most allergenic form of egg [18].

Regarding peanut and TN allergies, data from France are in the middle range. The HealthNuts Study found a higher prevalence of peanut allergy (1.9% at 4 years of age), but our results are in line with data from the Isle of Wight cohort (0.5% for peanut allergy and 0.2% for TN allergy at 4 years of age) and a US sample (0.2% in children aged 0–2 years) [16,18,19].

Regarding exotic fruits (kiwi, pineapple or banana), we found a high prevalence of FAs when compared with the previous data. When details on culprit exotic fruit were available, kiwi was the most frequently reported. Kiwi was previously reported as a common allergen in children in France [7,8] as was the case in Australia [24]. In Spain, kiwi was the most common allergen in “non-core food” after TNs [5].

When comparing the prevalence of each FA according to age, milk and egg allergies had a higher prevalence at younger ages, which suggests a high rate of resolution of these allergies in childhood [2,23]. In contrast, the prevalence of TN and peanut allergies was higher after 3 years of age, in concurrence with previous studies [16,19,25]. This finding could be due to an absence of known allergies at younger ages, because the introduction of TNs is usually delayed. In the HealthNut study, only 18.5% of children had already consumed TNs at 1 year of age. Moreover, peanut and TN allergies are known for their lower rate of resolution when compared with egg or milk allergies [26]. However, milk remains the most frequent allergy at 5.5 years of age, as in previous studies [27].

One-fifth of children with FAs had multiple-group FAs, which equates to 1.13% of the study population. This prevalence is similar to that found in the Isle of Wight cohort (0.7–1.3% of children with multiple FAs) [19], but is lower than the prevalence identified in a US cross-sectional survey (2.4%) [16]. An underestimation of prevalence is possible in the current study, because we considered groups of allergens to define multiple-group FAs. In the Isle of Wight cohort, cow’s milk and egg were the most common associated food allergens under 4 years of age, which were replaced by peanut and TN allergens in older children [19]. In the present cohort, peanut was often involved in multiple-group FAs, often in combination with egg and TN allergies. In 2018, McWilliam et al. estimated that 45% of 6-year-old children with a peanut allergy also had a TN allergy. Among children with egg and peanut allergies at 1 year, more than 35% had a TN allergy at 6 years of age [28]. In a cohort of 317 children with peanut allergy, Cousin et al. reported cross-allergy to TNs in 38.8% and to legumes in 7.9% of participants [29]. The MIRABEL study found a high rate of associated peanut and TN allergies [30]. The Pronuts study found a rate of 60.7% for coexistent peanut, TN and sesame seed allergy [25]. These findings highlight the need to screen associated allergies in children with FAs, in particular looking for TN and legume allergies in children with a peanut allergy.

As in other studies, FAs were frequently associated with other atopic diseases, such as asthma and ARC [6,23,31,32]. These atopic comorbidities must be screened for because asthma and ARC are particular risk factors for food anaphylaxis [3], and an altered skin barrier due to eczema leads to sensitization and FAs [31].

The ELFE study provides a unique opportunity to assess the prevalence of FAs in childhood. The prospective design limited memory bias, and a specific weighting was used in the assessment of prevalence to account for selection and attrition bias. The main limitation of the study is the parental reporting of diagnosis of allergies, because no details were available on any adverse reactions and tests leading to the diagnosis. Even though we tried to limit the bias by asking about food avoidance based on physician advice related to FAs, parental reporting could have led to an overestimation of FA prevalence. Therefore, our estimations are probably within the high range of the real prevalence of food allergies. To increase the validity of the definitions used in this study, reports of food intolerance, reactions to histamine-releasing foods and food avoidance not due to FAs were excluded. Moreover, the moderate agreement between parent- and physician-reported cases raises questions surrounding FA diagnosis and food avoidance conducted by parents. Based on a nationwide birth cohort, this study provides important French epidemiologic updated data that will prove essential for managing and preventing FAs.

## Figures and Tables

**Figure 1 nutrients-14-03624-f001:**
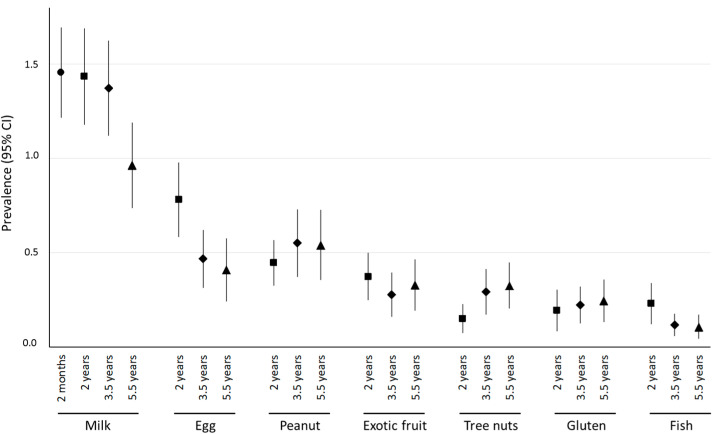
Prevalence of allergy to specific allergens at each time point (circle: 2 months (*n* = 15,785), square: 2 years (*n* = 12,537), diamond: 3.5 years (*n* = 11,389) and triangle: 5.5 years (*n* = 10,987) of age). CI: Confidence Interval.

**Figure 2 nutrients-14-03624-f002:**
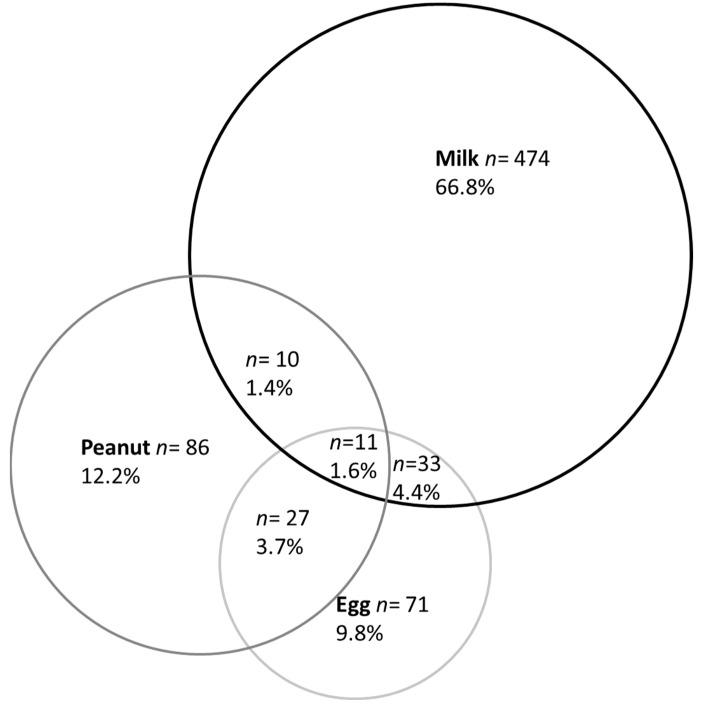
Venn diagram depicting the overlap of milk, egg and peanut allergies in children with food allergies up to 5.5 years of age. *n*: number.

**Figure 3 nutrients-14-03624-f003:**
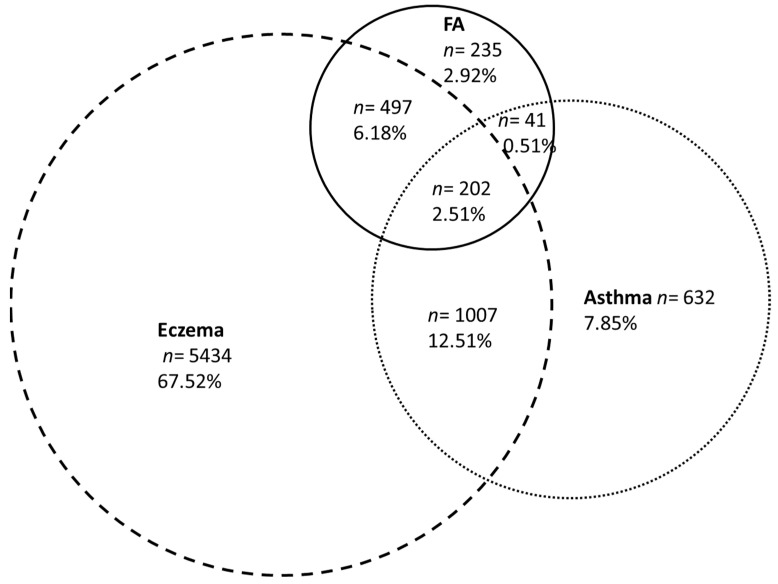
Venn diagram depicting the overlap of food allergies (FA), eczema and asthma in children with at least one allergic disease up to 5.5 years of age.

**Table 1 nutrients-14-03624-t001:** Allergen families.

Allergens Families	Food Avoidance Reported by the Parents in the Open-Ended Item	
Milk	Milk *, butter, cream, cheese, beef and veal allergies	
Egg		
Peanut		
Exotic fruits	Kiwi, pineapple, banana	
Fish		
Gluten		
Soy		
Tree nuts	Almond, cashew nut, pistachio, hazelnut, walnut, nut, chestnut, sesame, tree nut, coconut, macadamia nut	
Legumes	Peas, lentils, legumes, fava bean, chickpeas, flat bean, split bean	
Mustard		
Sea food	Sea food, shellfish, crab, shrimp	
Meat	Meat, chicken, turkey, lamb, pork, poultry, frog	
Vegetables	Eggplant, pepper, onion, yam, corn, zucchini, cabbage, artichoke, leek, mushroom, spinach, endive, vegetables, potato	
Pollen food allergies	Fruit (fig, mandarin, citrus, blueberry, stone fruit, blackcurrant, orange), apiaceae (parsley, celery, carrot), rosaceae (peach, apple, pear, mirabelle plum)	
Other	Food additives (coloring, aroma, preservatives, etc.), foods with several ingredients (sweetened beverages, fruits juice, spice, etc.), other foods (honey, vinegar, pickle, etc.)	

* We did not have details about the type of milk (cow, goat or sheep).

**Table 2 nutrients-14-03624-t002:** Sample characteristics (*n* * = 16,400).

		Weighted % (*n*)
Males		51.35% (8410)
Gestational age		
	Preterm (<37 weeks)	5.17% (866)
	Term (37 weeks or more)	94.83% (15,269)
C-section delivery		18.80% (3001)
Never breastfed		27.11% (4305)
Maternal age at delivery, years		
	<25	12.12% (1567)
	25–29	30.65% (4890)
	30–34	34.14% (6041)
	35 years or more	23.10% (3891)
Maternal education level		
	Up to lower secondary	5.33% (592)
	Upper secondary	40.52% (5330)
	Intermediate	20.40% (3553)
	3-year university degree	16.33% (2811)
	At least 5-year university degree	17.43% (3160)
Family history of atopy		48.09% (7878)

*n*: number

**Table 3 nutrients-14-03624-t003:** Prevalence of food allergy up to 5.5 years of age in the ELFE cohort (*n* = 16,400).

	Weighted Prevalence % (95%CI *)
At least one allergen	5.94 (5.54–6.34)
Milk	3.40 (3.09–3.71)
Egg	0.99 (0.80–1.18)
Peanut	0.93 (0.75–1.12)
Exotic fruits	0.65 (0.51–0.80)
Tree nuts	0.54 (0.40–0.68)
Gluten	0.41 (0.28–0.54)
Fish	0.37 (0.26–0.48)
Pollen-food allergies	0.28 (0.17–0.38)
Vegetables	0.19 (0.12–0.26)
Sea food/shellfish	0.25 (0.14–0.36)
Legumes	0.15 (0.08–0.22)
Mustard	0.12 (0.05–0.19)
Meat	0.08 (0.03–0.14)
Soy	0.08 (0.03–0.13)
Other	0.33 (0.22–0.45)

* CI: Confidence Interval

**Table 4 nutrients-14-03624-t004:** Allergens involved in single-group food allergies and multiple-group food allergies.

		Single-Group Food Allergy	Multiple-Group Food Allergies
Allergens			
	Milk (*n* = 578)	83.2%	16.8%
	Egg (*n* = 142)	35.9%	64.1%
	Peanut (*n* = 134)	41.0%	59.0%
	Exotic fruit (*n* = 101)	56.4%	43.6%
	Tree nuts (*n* = 80)	38.8%	61.3%
	Gluten (*n* = 56)	32.1%	67.9%
	Fish (*n* = 56)	41.1%	58.9%
	Vegetables (*n* = 34)	50.0%	50.0%
	Pollen food allergy syndrome (*n* = 35)	60.0%	40.0%
	Seafood (*n* = 26)	38.5%	61.5%
	Legumes (*n* = 23)	43.5%	56.5%
	Mustard (*n* = 15)	13.3%	86.7%
	Soy (*n* = 14)	7.1%	92.9%
	Meat (*n* = 13)	30.8%	69.2%
	Others (*n* = 45)	64.4%	35.6%
Comorbidities			
	Eczema	68.5%	78.5%
	Asthma	22.1%	35.4%
	Allergic rhinitis or conjunctivitis	39.4%	52.7%

## Data Availability

The data underlying the findings cannot be made freely available for ethical and legal restrictions. This study includes a substantial number of variables that, together with the data used, could be employed to re-identify participants based on a few key characteristics, which could then provide access to other personal data. Therefore, the French ethics authority strictly forbids making these data freely available. However, some data can be obtained upon request from the ELFE principal investigator. Readers may contact marie-aline.charles@inserm.fr to request the data.

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
