# Peer review of "Prevalence of Food Allergy in France up to 5.5 Years of Age: Results from the ELFE Cohort"

_nutrients, 2022, doi:10.3390/nu14173624_

Round 1

Reviewer 1 Report

The focus of the study is to evaluate the prevalence of food allergy (FA) based on the data collected from ELFE study. The follow up study obtained from the phone interview of parents with children within 5.5 years which born in mainland France in year 2011. After reading the manuscript there are some points that have called my attention and, therefore, I have a few comments for the authors. 

The pollen food allergies group contains many types of fruits. Although the prevalence is 0.28 shown in table 3, the single food allergy association is quiet high to 60%. Authors might want to clarify them compared to exotic fruits in detail.

In figure 1, the sharply decreased of Milk allergen at 5.5 yrs, but at 3.5 yrs of Egg, and then partly increased of Peanut at 3.5 to 5.5 yrs. However, the associated results of these three allergies in figure2 shows very less in milk with egg. Although these Venn diagram demonstrated FA children which age 5.5 years, the specific neck years need to separately depict at Venn diagram.

The prevalence of allergy disease in this cohort using children count as N that born in 2011 without comparing to other years. Also, the FA is the main topic of this manuscript, but the frequency of FA is less than AD and asthma which shown in figure 3. Thus hard to conclude co-disease happens in FA with other allergy diseases. Although some confounding factors has been discussed in discussion.

Author Response

The focus of the study is to evaluate the prevalence of food allergy (FA) based on the data collected from ELFE study. The follow up study obtained from the phone interview of parents with children within 5.5 years which born in mainland France in year 2011. After reading the manuscript there are some points that have called my attention and, therefore, I have a few comments for the authors.

Response: Thank you for your comments and for giving us the opportunity to improve our manuscript.

 The pollen food allergies group contains many types of fruits. Although the prevalence is 0.28 shown in table 3, the single food allergy association is quiet high to 60%. Authors might want to clarify them compared to exotic fruits in detail.

Response: Thank you for this accurate comment. Indeed, the group "pollen food allergies" contains several types of fruits and most of children (60%) having a FA to this group do not have food allergy to other food groups. These food allergies are frequently associated with aeroallergen allergy but less frequently with other food allergies. From your comment, we understand that we needed to clarify our definition of multiple food allergy. So we changed "single (or multiple) food allergy" for "single- (or multiple-) group food allergy" throughout the text.

 Regarding exotic fruits, we choose to separate them from other fruits as in previous studies (references: 5,7,19) because, contrary to fruit included in the “pollen food allergy group”, a food allergy to exotic fruits is more often an authentic food allergy and not just a crossed allergy. We had the details of culprit exotic fruits only if the parents had answered the open-ended question by kiwi, pineapple or banana (see Table 1).

In figure 1, the sharply decreased of Milk allergen at 5.5 yrs, but at 3.5 yrs of Egg, and then partly increased of Peanut at 3.5 to 5.5 yrs. However, the associated results of these three allergies in figure2 shows very less in milk with egg. Although these Venn diagram demonstrated FA children which age 5.5 years, the specific neck years need to separately depict at Venn diagram.

Response: The primary objective of our study was to describe the global prevalence of food allergy among preschool children and to detail the main food allergens involved. Figure 1 presents the time point prevalence at each time point and ewe added this precision on the title of Figure 1 (line 171). As our questionnaire did not allow to collect cured food allergies at each follow-up, we did not focus on the evolution of food allergies over time, even if it would be of great interest in studies with relevant data. That’s why the Venn diagrams were presented globally from birth to 5.5 years and not at each time point.

 The prevalence of allergy disease in this cohort using children count as N that born in 2011 without comparing to other years. Also, the FA is the main topic of this manuscript, but the frequency of FA is less than AD and asthma which shown in figure 3. Thus hard to conclude co-disease happens in FA with other allergy diseases. Although some confounding factors has been discussed in discussion.

Response: Thank you for your comment. All children in the ELFE cohort were born in 2011 so the food allergy prevalence is estimated only in children born in 2011 and there is no similar data available in France or in Europe in the past 15 years to make a more accurate comparison.

As previously described in other studies, the prevalence of asthma and eczema were higher than the prevalence of food allergies. However, as shown in figure 3, most children with a food allergy had at least one other atopic disease. It highlights the need for a global assessment and management of allergic diseases in case of food allergy. As our study is descriptive, we did not study causality or risk factors for food allergies. We have only described the prevalence of each atopic disease in children with food allergies and vice versa. That is why we did not take into account any confounding factor. To clarify this point, we changed the sentence line 273 (marked version) in the discussion and modified the title of figure 3.

Reviewer 2 Report

In this study, the authors investigated the prevalence of FAs and the most common allergens in a French nationwide birth cohort. They found FAs were reported for 5.94% children and milk was the most common allergen. They also found about 20% children with allergies had multiple FAs. In general, this study has certain merits including a relatively large population and a long-term follow-up period, which is of some interests to readers. However, there are some questions to be addressed.

1.      The data of FA diagnosis was based on parent reports, which might have a high risk of overestimate the prevalence of FAs. The parents had difficult to distinguish FAs from other adverse food reactions.

2.      The detailed information of FA diagnosis procedure should be clarified in the methods, as it was the most important element for the study. What kind of information would be collected to diagnose FAs?

3.      The diagnosis criteria for other allergic diseases were too simple. For example, did the authors think atopic dermatitis can be diagnosed only by an itchy rash?

4.      the authors may need to present the prevalence data of parent-reported FAs and physician-diagnosed FAs and the differences can be further discussed.

5.      The limitation of the study should be addressed adequately in the discussion and need to note that the results should be interpreted with caution.  

Author Response

In this study, the authors investigated the prevalence of FAs and the most common allergens in a French nationwide birth cohort. They found FAs were reported for 5.94% children and milk was the most common allergen. They also found about 20% children with allergies had multiple FAs. In general, this study has certain merits including a relatively large population and a long-term follow-up period, which is of some interests to readers. However, there are some questions to be addressed.

Thank you for your kind comments and your accurate suggestions.

  1. The data of FA diagnosis was based on parent reports, which might have a high risk of overestimate the prevalence of FAs. The parents had difficult to distinguish FAs from other adverse food reactions.

Response: We acknowledge that our definition of food allergy based on parental report of food avoidance on medical advice due to a food allergy may lead to an overestimation of the prevalence of food allergies. This limitation was addressed in lines 279-282 of the discussion. We added a sentence in the discussion (lines 282-283 of the marked version of the manuscript) in order to be specific regarding this risk of overestimation. We hope that this change will answer your comment. We also wanted to highlight that in the ELFE questionnaires, the question regarding food allergy is ”Does your child avoid a food on medical advice because of a food allergy ?” (lines 74-75).  Asking about food avoidance based on physician advice due to FAs aimed to limit the risk of parents reporting suspected allergies not discussed with their doctor. In order to clarify that we describe parent-reported food allergies, we added the information in the conclusion (line 290).

  1. The detailed information of FA diagnosis procedure should be clarified in the methods, as it was the most important element for the study. What kind of information would be collected to diagnose FAs?

Response: The methodology regarding information collected for food allergies is described lines 73 to 82. Parents were asked to report all food avoidance on medical advice due to food allergy. Apart from the 2-yr physician questionnaire, we did not collect data regarding sensitization or oral food challenge. In order to clarify this point, we added the following sentence line 82-83 of the marked version of the manuscript “No information regarding symptoms or diagnosis procedure was collected from the parents.”. In the discussion, we highlighted that different diagnostic criteria can induce variations in FA prevalence estimation (lines 220-223).

  1. The diagnosis criteria for other allergic diseases were too simple. For example, did the authors think atopic dermatitis can be diagnosed only by an itchy rash?

Response: We acknowledge that the diagnostic criteria for allergic diseases were quite simple and added more detail on collected data in the revised version of the manuscript. Regarding the definition of atopic dermatitis in the ELFE cohort, parents were asked if their child had an itchy rash at the 1-yr follow-up but at 2, 3.5 and 5.5years old, the parents were asked to report medical diagnosis of eczema (and not itchy rash). We added this precision line 96-97 of the marked version of the manuscript. As suggested in your comment, atopic dermatitis is too specific to be used in our study. Thus, we changed the term "atopic dermatitis" to "eczema" throughout the text.

  1. the authors may need to present the prevalence data of parent-reported FAs and physician-diagnosed FAs and the differences can be further discussed.

Response: Thank you for your suggestion. As data on food allergies reported by the physician are only available at 2 years old, we can compare physician-reported prevalence with parent-reported prevalence only at 2 years of age. At this time point, the prevalence of physician-reported food allergies was 3.14% versus 3.40% for the parent-reported prevalence. We added the 2-year parent-reported prevalence of food allergy in the first sentence of the paragraph regarding physician-reported food allergy (line 188 of the marked version of the manuscript). However, even if prevalence seems to be similar, agreement was moderate.

  1. The limitation of the study should be addressed adequately in the discussion and need to note that the results should be interpreted with caution.

Response: Thank you for your comment. We specified in the discussion that parental report of food allergies could lead to an overestimation of the prevalence of food allergies (lines 282-284). In the conclusion, we added the precision that we described the prevalence of food allergies according to parental report (line 289). We hope that these changes will answer your comments.

Round 2

Reviewer 2 Report

The authors have answered the questions adequately. Although there are some limitations in this study, it is still of interests to readers. I have no further comments.